# Phase transitions in 2D multistable mechanical metamaterials via collisions of soliton-like pulses

Weijian Jiao [1,2], Hang Shu[1], Vincent Tournat [3], Hiromi Yasuda[1,4,5] & Jordan R. Raney [1] ✉

In recent years, mechanical metamaterials have been developed that support the propagation of an intriguing variety of nonlinear waves, including transition waves and vector solitons (solitons with coupling between multiple degrees of freedom). Here we report observations of phase transitions in 2D multistable mechanical metamaterials that are initiated by collisions of soliton-like pulses in the metamaterial. Analogous to first-order phase transitions in crystalline solids, we observe that the multistable metamaterials support phase transitions if the new phase meets or exceeds a critical nucleus size. If this criterion is met, the new phase subsequently propagates in the form of transition waves, converting the rest of the metamaterial to the new phase. More interestingly, we numerically show, using an experimentally validated model, that the critical nucleus can be formed via collisions of soliton-like pulses. Moreover, the rich direction-dependent behavior of the nonlinear pulses enables control of the location of nucleation and the spatio-temporal shape of the growing phase.

Nonlinear mechanical metamaterials have received significant attention in the past decade, due to their versatile static and dynamic behaviors[1,2], and the ability to tune their response[3,4]. For example, nonlinear mechanical metamaterials have been previously designed that exhibit tunable kinematics[5], stiffness[6], Poisson's ratio[7], thermal expansion[8], and band gaps[9,10]. Nonlinear mechanical metamaterials often exhibit rich amplitude-dependent properties, such as weakly nonlinear harmonic waves[11–13], cnoidal waves[14], solitons[15,16], and transition waves[17–24].

One particular class of mechanical metamaterial obtains its nonlinear properties from the rotation of periodic internal features, such as squares connected at their hinges. Systems based on the rotating-squares mechanism have long been studied due to their interesting static properties (i.e., their auxetic characteristics)[25–27]. More recently, it has been observed that they are also capable of propagating a variety

of nonlinear waves[14,28–31]. A notable example is the propagation of vector solitons, which have coupled translational and rotational degrees of freedom (DOFs) and can display distinct solitary modes for different propagation directions[32]. Interactions of these nonlinear waves have also been investigated, albeit mostly for one-dimensional systems[33]. Due to the coupling between different DOFs, which is less often considered in Hertzian granular media[34–38], the collision of vector solitons has been shown to exhibit anomalous phenomena, including repelling, destruction, etc., in addition to classical soliton collisions[39].

Recently, the dynamics of multistable versions of these systems have also been studied. For example, multistability can be achieved by introducing permanent magnets[29,40], which can produce multiple energy minima, each associated with equilibrium angles that the squares can snap between. If squares are rotated from one stable angle to another, it is possible for this reconfiguration to propagate

[1]Department of Mechanical Engineering and Applied Mechanics, University of Pennsylvania, Philadelphia, PA, USA. [2]School of Aerospace Engineering and Applied Mechanics, Tongji University, Shanghai, China. [3]Laboratoire d'Acoustique de l'Université du Mans (LAUM), UMR 6613, Institut d'Acoustique - Graduate School (IA-GS), CNRS, Le Mans Université, Le Mans, France. [4]Aviation Technology Directorate, Japan Aerospace Exploration Agency, Mitaka, Tokyo, Japan. [5]Institute of Space and Astronautical Science, Japan Aerospace Exploration Agency, Sagamihara, Kanagawa, Japan. ✉e-mail: raney@seas.upenn.edu

throughout the structure in the form of a transition wave. In addition, the collision of transition waves of incompatible type can cause the formation of stationary domain walls, which can be exploited for the design of reconfigurable metamaterials[29].

Here, we investigate collisions of nonlinear, soliton-like pulses in 2D multistable systems of rotating squares, and how these collisions can be used to remotely nucleate phase transitions at arbitrary locations. As a first step, we experimentally and numerically show how phase transitions can be initiated via quasistatic rotation of a "critical nucleus" of squares, analogous to nucleation during first-order phase transitions[41,42]. Note, that in this work, the phase transitions are enabled by multistability, which is achieved by embedding magnets in the squares. This is in contrast with other work[43–45], in which phase transitions are induced by applying static precompression to the entire system, or by dynamic recoil[46]. Second, we investigate the criteria necessary for collisions of soliton-like pulses to induce this phase transition. Finally, we describe how the anisotropy associated with the symmetry of the system produces direction-dependent nucleation and propagation of the phase transition. The presented method for nucleation of phase transitions could enable new insights for the design of high-dimensional reconfigurable, shape-transforming, and deployable mechanical metamaterials. For example, a deployable structure can be designed to exhibit monostability or multistability at arbitrary locations, and the phase (shape) of the multistable parts, even if they are located in the bulk, can still be controlled (possibly independently) by exciting pulses from the boundary.

## Results
### Phase transitions in multistable metamaterials

We start by experimentally and analytically characterizing the energy landscape of the building block of the mechanical system, i.e., a $2 \times 2$ set of squares. To experimentally measure the behavior of such a system, we fabricate an elastomeric building block, following a

conventional molding-casting process. Specifically, we design and 3D print a mold (MakerGear M2, polylactic acid). We then pour silicone precursor (Dragon Skin 10) into the mold and allow it to cure. The squares have side length 12 mm and are connected by thin hinges of thickness 1.5 mm. Permanent magnets are inserted into each square (see Methods for fabrication details). A schematic of the building block is shown in Fig. 1a. The competition between the strain energy of the hinge and the interaction of the magnets gives the squares three stable angles[29]. Each of these corresponds to a local minimum in the potential energy landscape (Fig. 1b). Then, to quantify the effects of different design parameters, we introduce a discrete model capable of capturing the multistable energy landscape. Each square, assumed to be a rigid body with mass $M$ and moment of inertia $J$ (experimentally measured to be $M = 2.501$ g and $J = 60.024 \times 10^{-9}$ kg·m²), has two translational degrees of freedom ($u$ and $v$) and one rotational degree of freedom ($\theta$). Each hinge is modeled by three springs (Fig. 1a): a linear longitudinal spring with stiffness $K_l$, a linear shear spring with stiffness $K_s$, and a nonlinear torsional spring with potential energy $E_\theta(\Delta\theta)$ expressed as

$$E_\theta(\Delta\theta) = \frac{1}{2}K_\theta(\Delta\theta)^2 + V_{\text{Morse}}(\Delta\theta), \tag{1}$$

$$V_{\text{Morse}}(\Delta\theta) = A\left[e^{2\alpha(\Delta\theta + 2\theta_0 - 2\theta_M)} - 2e^{\alpha(\Delta\theta + 2\theta_0 - 2\theta_M)}\right] + A\left[e^{-2\alpha(\Delta\theta + 2\theta_0 + 2\theta_M)} - 2e^{-\alpha(\Delta\theta + 2\theta_0 + 2\theta_M)}\right], \tag{2}$$

where $K_\theta$ is the linear torsional spring constant, $\theta_0$ is the initial equilibrium angle, $\Delta\theta$ is the relative angle of the hinge, and $V_{\text{Morse}}$ is the Morse potential, which is used to empirically describe the nonlinear magnetic interactions between squares. In Eq. (2), $A$ and $\alpha$ define the depth and width of the Morse potential, respectively, and $\theta_M$ determines the equilibrium points. To obtain these parameters for the

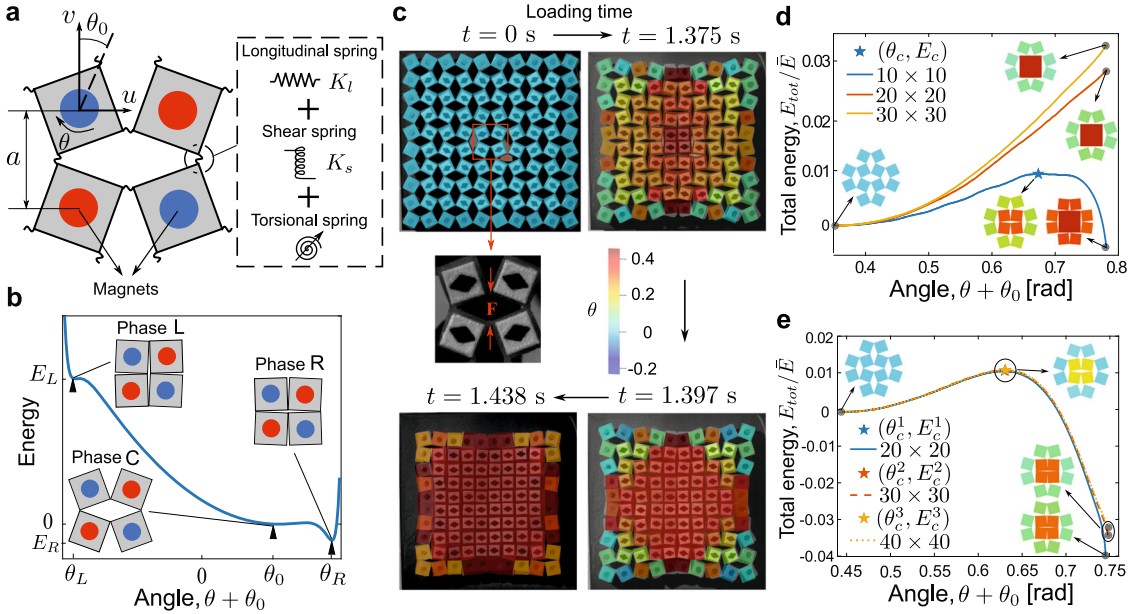

**Fig. 1 | Experimental observations and numerical characterization of quasistatically-induced phase transitions in 2D multistable metamaterials.** **a** Building block of the metamaterial. **b** Multistable potential energy landscape of the hinge. **c** Optical images of an experimental specimen with the center four squares subjected to quasistatic rotation via the application of the force $F$; this causes the formation of a new phase and its eventual growth outward through the rest of the metamaterial. The angles obtained by the numerical simulations are superimposed over the experimental images (indicated by color). The positive (negative) rotational direction is defined in a way that rotates the squares from the

initial Phase C to the new Phase R (Phase L). **d** Effect of system size on the formation of a $2 \times 2$ nucleus, using the validated numerical model with experimentally-obtained parameters ($K_l \approx 3.958 \times 10^2$ N/m, $K_s \approx 58$ N/m, $K_j \approx 2.5 \times 10^{-4}$ N·m/rad, $A \approx 2 \times 10^{-4}$, $\alpha \approx 8.5$, $\theta_0 \approx 20°$, $\theta_M \approx 45°$). **e** New parameters were selected and normalized for the subsequent numerical simulations of this study, ensuring that a critical nucleus size of $2 \times 2$ squares would be obtained ($K_1 = 0.2$, $K_2 = 0.0306$, $\beta = 3.0556$, $\bar{A} = 0.0186$, $\alpha = 11$, $\theta_0 = 25.4°$, $\theta_M = 43°$). In both **d** and **e**, the total energy (normalized by $\bar{E} = K_l a^2$) is that of the 12 squares, including the center $2 \times 2$ squares and their eight neighbors for three different system sizes.

numerical simulations, we conduct experimental tensile and shear tests using a commercial quasistatic test system (Instron model 68SC-5) with custom fixtures (Supplementary Figs. 2 and 3). These are designed to allow the squares to rotate during the tests. Then, Eq. (1) gives the energy landscape of the building block, which exhibits three distinct phases (labeled as Phase L, C, and R), as shown in the inset of Fig. 1b.

Next, we focus on the dynamic behavior of the multistable system. We derive the equations of motion (EOMs) of the system. By introducing the following normalized parameters: $K_1 = K_s/K_l$, $K_2 = K_\theta/(K_l a^2)$, $T = t\sqrt{K_l/M}$, $\beta = a\sqrt{M/J}$, $\bar{A} = A/(K_l a^2)$, $U = u/a$, $V = v/a$ (where $a$ is the distance between the centers of two neighboring squares), we obtain the dimensionless EOMs (see Supplementary Note 1). Then, we investigate the dynamic response of the system by numerically solving the EOMs, using the fourth-order Runge-Kutta method.

Before performing numerical simulations to explore whether collisions of impulses can induce a phase transition, we first conduct experiments to validate the discrete model. To this end, we fabricate a larger prototype of size $10 \times 10$ squares, following the same procedures described earlier (note, to reduce the effect of the boundaries on the behavior of the mechanical system, magnets are not embedded in the exterior squares). The system is initially in Phase C. Nucleation is induced by quasistatically forcing the $2 \times 2$ squares at the center of the specimen to the new Phase R. Once the new phase is nucleated, the entire specimen subsequently undergoes a transition to the new phase, as shown in the optical images of Fig. 1c, obtained via a high-speed camera (Photron FASTCAM Mini AX; Supplementary Movie 1 and Supplementary Figs. 4 and 5).

We perform numerical simulations by numerically solving the EOMs of a system of $10 \times 10$ squares, with the parameters obtained from experimental tensile and shear tests (see Methods for details): $K_l \approx 3.958 \times 10^2$ N/m, $K_s \approx 58$ N/m, $K_j \approx 2.5 \times 10^{-4}$ N·m/rad, $A \approx 2 \times 10^{-4}$, $\alpha \approx 8.5$, $\theta_0 \approx 20°$, and $\theta_M \approx 45°$. In Fig. 1c, we superimpose the angles predicted by the simulations (indicated by color) on the experimental images, showing excellent agreement.

We now rely on numerical simulations to characterize some fundamental properties of the phase transition, such as the critical nucleus size and the energy threshold that quantifies the minimum amount of energy required for the phase transition to occur in an ideal, thermodynamic sense. Analogous to classical first-order phase transitions, we define the critical nucleus size for our system as the minimum square size in a new phase from which the new phase is stable and begins to grow. Here, we intentionally aim to produce the smallest critical square nucleus, i.e., $2 \times 2$ squares, to make it easier to induce phase transitions via collisions. To identify the energy threshold for the quasistatically induced phase transition, we calculate the total energy (normalized by $\bar{E} = K_l a^2$) of the 12 squares, including the center four squares (where the loading is applied) and their nearest neighbors, as a function of angle $\theta_0 + \theta_{in}$ ($\theta_{in}$ is the input rotation induced by the quasistatic loading). The total energy can be divided into two components. The first is associated with the center four squares (i.e., energy change as they move from Phase C to Phase R). The second is the amount of energy associated with the interface (i.e., the eight nearest neighbors of the center four squares). Interestingly, we observe that the total energy displays an energy barrier $E_c$ at $\theta_0 + \theta_{in} = \theta_c$, as indicated by the blue star in Fig. 1d. The existence of an energy barrier implies that, under certain input rotations, the phase transition may be favourable once the energy barrier is overcome during quasistatic loading.

To investigate the effects of system size on nucleation, we repeat the above analysis for two larger systems of size $20 \times 20$ and $30 \times 30$. As shown by the yellow and red curves in Fig. 1d, the total energy monotonically increases as the center four squares rotate to Phase R. The interface energy (the second energy component) surpasses the energy released from the phase change (the first energy component),

which indicates that a nucleus size of $2 \times 2$ squares is not big enough to start the nucleation process (and therefore the growth of the new phase is not favourable). The critical nucleus size is a thermodynamic quantity that does not depend on the system size. Thus, by definition, a $2 \times 2$ nucleus size cannot be identified as the critical nucleus size for systems with the experimentally-obtained parameters, even though it can trigger a phase transition in the $10 \times 10$ system (due to boundary effects).

To achieve a critical nucleus size of $2 \times 2$ squares, we choose the following parameters: $\theta_0 \approx 25.4°$, $\theta_M = 43°$, $K_1 = 0.2$, $K_2 = 0.0306$, $A = 2.74 \times 10^{-3}$, $\alpha = 11$, and $\beta = 3.0556$. With these parameters, we perform an energy analysis for three systems of size $20 \times 20$, $30 \times 30$, and $40 \times 40$ (Fig. 1e). The three energy curves overlap for the most part, with three nearly identical energy barriers $E_c \approx 1.064 \times 10^{-2}$, and only differ slightly as the loading angle gets close to Phase R. These findings indicate that, for the given parameters, the effects of system size on the nucleation is negligible for system size greater than $20 \times 20$. Thus, we have achieved a critical nucleus of $2 \times 2$ squares. For completeness, we report in Supplementary Fig. 6 snapshots from the numerical simulation for a system of $30 \times 30$ squares, exhibiting the nucleation of a critical nucleus of $2 \times 2$ squares and the phase transition that propagates outward throughout the rest of the metamaterial in the form of a transition wave (see Supplementary Movie 2 and characterization of the transition wave in Supplementary Fig. 7). Finally, we note that it is possible to obtain a critical nucleus size other than $2 \times 2$ squares (Supplementary Fig. 8).

## Initiating phase transitions via collisions of soliton-like pulses

Now that we have chosen parameters that produce the smallest critical square nucleus (i.e., $2 \times 2$ squares), we next consider how such a critical nucleus could be nucleated by colliding vector solitons. Previous studies[28,32] have shown that vector solitons can propagate in similar but monostable systems of squares without magnets. We first ask whether the multistable metamaterial can support the propagation of vector solitons or soliton-like pulses. To derive analytical solutions for solitons, we develop a continuum model by taking the continuum approximation of the discrete model and fitting a polynomial to the torsional potential within the energy well around the initial equilibrium $\theta_0$ (see derivation in Supplementary Note 2). Using the analytical findings as guidelines, we perform full-scale simulations to explore the potential of using collisions of soliton-like pulses to initiate a phase transition.

Here, we consider a nearly circular system with 30 squares along its diagonal. We impact the sample at different squares along its circumference to initiate pulses that propagate along different directions. Specifically, the impacts are displacement profiles in the form

$$D(T) = \frac{A_0}{2}\tanh\left[(T - T_0)/W\right] + \frac{A_0}{2}\tanh(T_0/W) \tag{3}$$

where $A_0$ and $W$ are parameters that alter the impact amplitude and shape, respectively. To avoid triggering a nucleation directly at the impacted squares, in the simulations we impose $\theta = 0$ to all squares on the boundary.

## Head-on collisions of two pulses with the same rotation

We first investigate head-on collisions of pulses by applying impacts at the left and right boundary. In Fig. 2a, we show snapshots of the wavefields at $T = 15$, 28.7, 35, and 60, demonstrating that a phase transition is induced where the two pulses collide (Supplementary Movie 3). By sweeping the impact amplitude $A_0$, we identify a critical amplitude $A_c = 0.306$, below which a nucleation is not induced by the colliding pulses (see Fig. 3a and Supplementary Movie 4). When $A_0 \geq A_c$, the collision of the two pulses can lead to the formation of a critical nucleus. In that case, the new phase propagates outward to the rest of

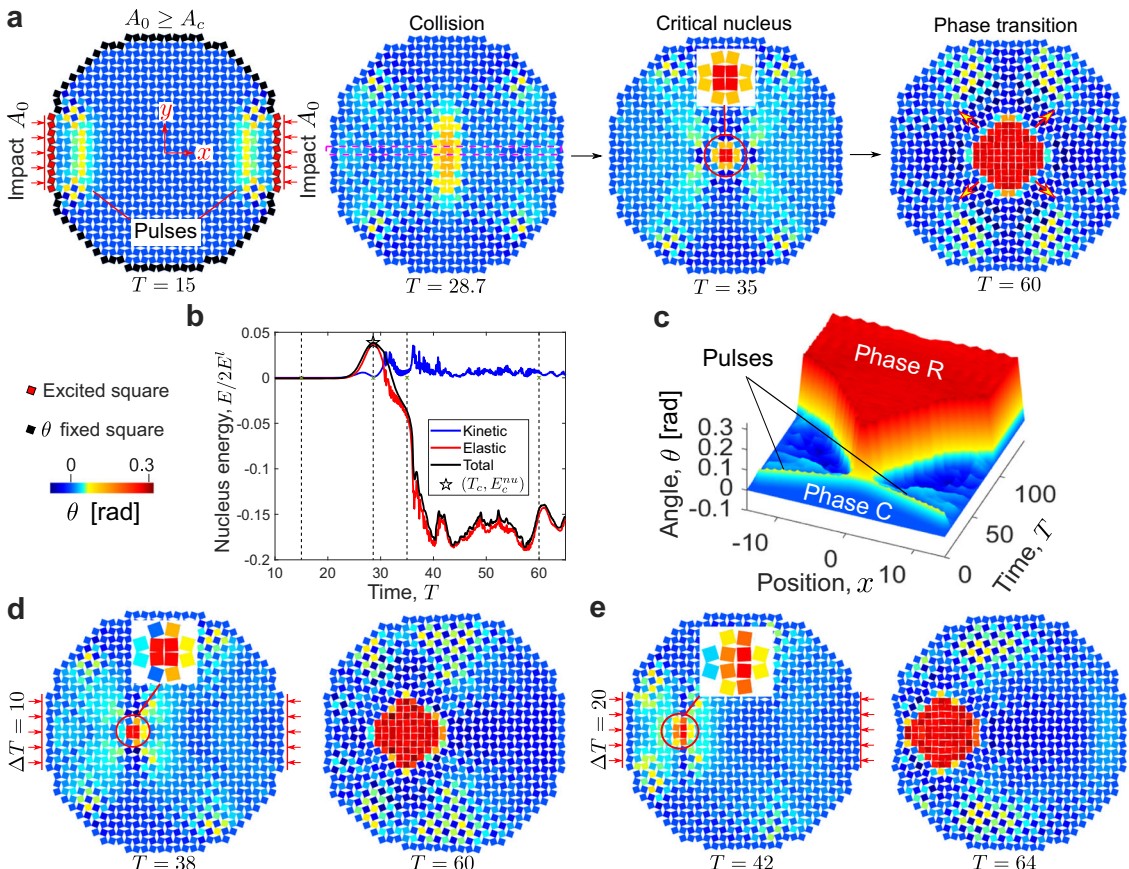

**Fig. 2 | Head-on collisions of two soliton-like pulses with nucleation of phase transitions. a** Snapshots of wavefields for $A_0 = 0.306 \equiv A_c$: before collision at $T = 15$, during collision at $T = 28.7$, nucleation at $T = 35$, and phase transition at $T = 100$ ($T = t\sqrt{K_t/M}$ is the normalized time). **b** Normalized energy of the cluster at the nucleation site as a function of time, suggesting an energy barrier $E_c^{nu}$ in the total energy curve. **c** Spatiotemporal plot obtained from the numerical simulation, showing the angle $\theta$ for squares along the propagation direction ($x$ axis) as a function of time. **d** and **e** Control of the location of nucleation via timing of the impulses for **d** $\Delta T = 10$ and **e** $\Delta T = 20$, where $\Delta T$ is the time delay of the impact on the left boundary with respect to the impact on the right boundary.

the structure via a transition wave. In Fig. 2b, we plot the normalized energy of the squares at the nucleation site (i.e., the squares in the inset of the third snapshot in Fig. 2a) as a function of time for $A_0 = A_c$. We observe that there also exists an energy threshold $E_c^{nu} = 3.84 \times 10^{-2}$ during the collision process. Comparing this energy threshold $E_c^{nu}$ with its counterpart in the previous quasistatic analysis, we note that $E_c^{nu}$ is much larger than $E_c$, a result of the fact that not all of the energy in the propagating pulses will be directed toward forming a new phase during the collision (e.g., some energy is lost in the form of scattered waves). Figure 2c shows a spatiotemporal plot that provides the angle of the squares along the propagation direction ($x$ axis) as a function of time and position (see Fig. 3b for a case of $A_0 < A_c$). We also note that the location of nucleation can be changed simply by introducing a time delay $\Delta T$ for the initiation of the impulse on the left with respect to the initiation of the impulse on the right. We demonstrate this by showing snapshots of the simulations in Fig. 2d and e for $\Delta T = 10$ and 20 (Supplementary Movie 5), respectively.

### Head-on collisions of two pulses with the opposite rotation
We also explore head-on collisions of pulses with different rotational directions (Fig. 3). In contrast with collisions between impulses with the same (positive) rotation (as was triggered by applying two compressive impulses at the left and right boundaries in Fig. 2a), Fig. 3c shows a collision of two pulses with opposite rotational directions, whose spatiotemporal plot is given in Fig. 3d. This is accomplished by changing the excitation at the right boundary from a compressive impact to a tensile impact. The two pulses pass through each other

without inducing a nucleation for $A_0 = A_c$ (Supplementary Movie 6). To better understand this observation, we separate the kinetic energy into two components: one associated with translational motion and the other associated with rotational motion. The results are plotted in Fig. 3e and f with $A_0 = A_c$ for the same rotation and opposite rotation, respectively. We find that there is some energy exchange between the two kinetic energy components for the same rotation case, i.e., some portion of the translational kinetic energy is transferred to the rotational kinetic energy, which is associated with the change in relative angle between squares. However, this energy exchange is almost negligible for the opposite rotation case. This is because, by our definition of the positive/negative rotational direction, the change of the relative angle between two connecting squares is less significant (or even zero) when the two squares have rotations of opposite direction (and of the same amplitude). This finding implies that the rotational kinetic energy gained during the collision process is critical for overcoming the energy barrier associated with nucleation. Another interesting scenario is the collision of two pulses with negative rotation, triggered by two tensile impulses. In this case, the energy exchange is almost negligible. As a result, nucleation cannot be initiated (Supplementary Fig. 9).

### Effects of propagation distance on nucleation
Since the pulses are triggered at the boundary and collide at the center of the structure, it is expected that the propagation distance can affect the wave interactions during the collisions, and therefore may affect the nucleation. We repeat the above analysis for systems with different

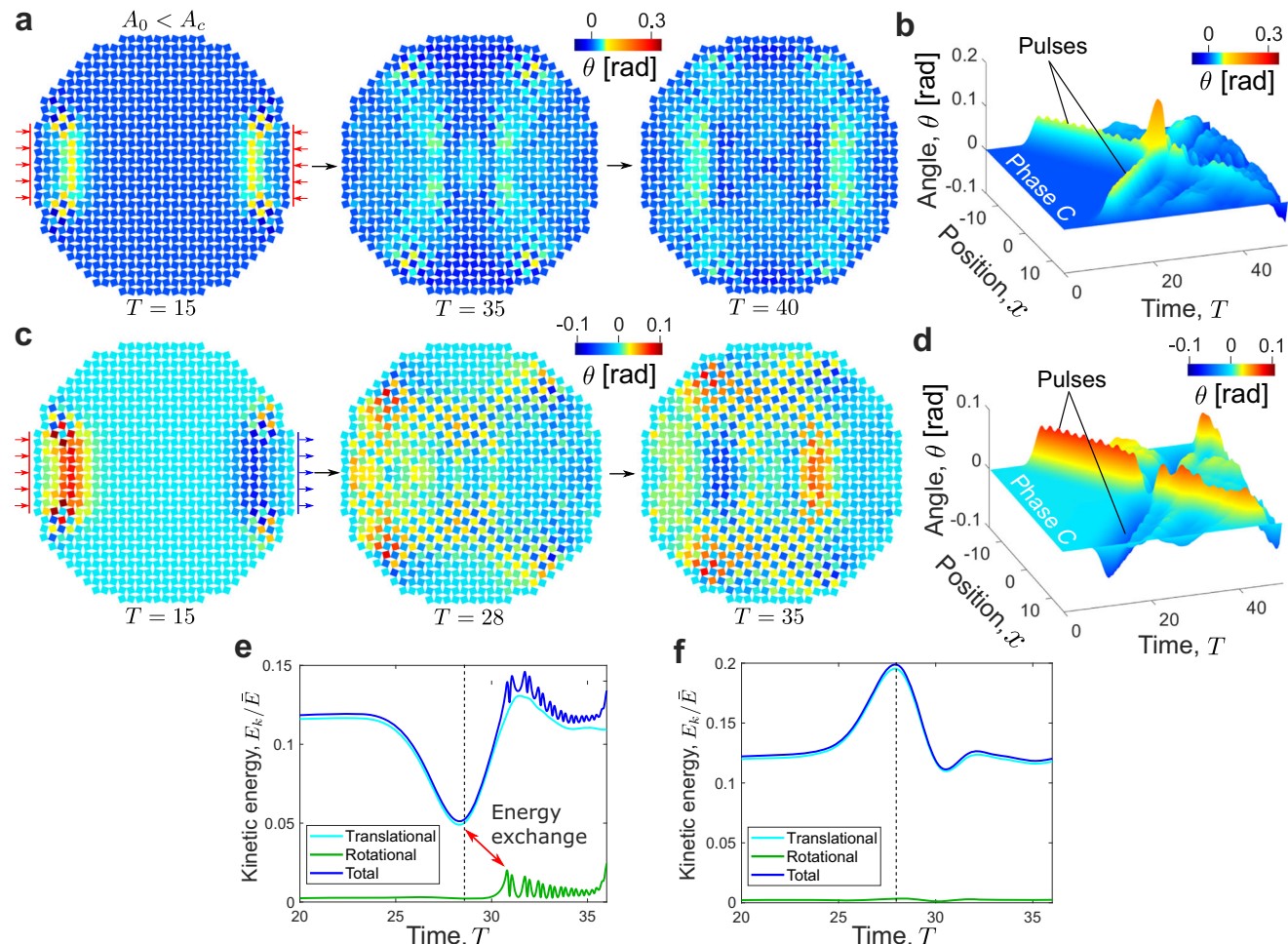

**Fig. 3 | Head-on collisions of two soliton-like pulses without nucleation of phase transitions. a** Head-on collision of two pulses with the same rotational direction for $A_0 = 0.3 < A_c$: snapshots of wavefields before collision at $T = 15$ and after collision at $T = 35$ and $40$, resulting in no phase transition. **b** Spatiotemporal plot extracted from the numerical simulation, showing the angle $\theta$ for squares along the propagation direction as a function of time. **c** Head-on collision of two pulses with the opposite rotational direction for $A_0 = A_c$: snapshots of wavefields before collision at $T = 15$ and after collision at $T = 28$ and $35$, showing that the pulses pass through each other. **d** Spatiotemporal plot obtained from the numerical simulation, showing the angle $\theta$ for squares along the propagation direction as a function of time. **e** and **f** Normalized kinetic energy of the whole structure as a function of time for a head-on collision of pulses for $A_0 = A_c$ with **e** the same rotational direction and **f** the opposite rotational direction (the vertical dashed lines indicate the time when the two pulses collide); the former case exhibits a significant exchange between the translational and rotational components of the kinetic energy.

sizes to examine this effect. The results, as reported in Supplementary Figure 10, show that the critical amplitude $A_c$ increases significantly as the size increases. We observe dispersion, especially in the direction perpendicular to propagation, which is qualitatively similar to the expected 2D dispersion behavior observed previously[32]. As a result, its amplitude spatially decays as it propagates through the media. In contrast, the critical energy barrier $E_c^{nu}$ does not change in an appreciable way, which indicates that the energy barrier for inducing a nucleation is a local quantity, and therefore there is no statistically significant change to the energy barrier.

## Collisions of pulses at other angles

Finally, we consider the effect of propagation direction on the ability of colliding pulses to nucleate a new phase (Fig. 4 and Supplementary Movie 7). The circular shape of the system allows facile excitation of pulses with arbitrary directions. For example, by applying impacts at the left and top boundary, the two pulses can propagate along both the $x$ and $y$ principal axes (i.e., the positive $x$ direction and the negative $y$ direction, respectively). As shown in Fig. 4b, the two pulses nucleate a new phase during their collision. In this case, the nucleation can be

induced at impact amplitude $A_0 = 0.292$, which is lower than the critical amplitude of a head-on collision (replotted in Fig. 4a). In addition, we observe that, after nucleation, the new phase grows predominantly along the diagonal, at $45°$ relative to the $x$ and $y$ axes. We refer to such pulses, traveling along the $x$ or $y$ axes, as mode I pulses. Another feasible propagation direction is along the diagonals (referred to as mode II pulses), a direction previously found to support the propagation of vector solitons in monostable systems of rotating squares[32]. Figure 4c shows a head-on collision between impulses propagating along this direction. Mode-I pulses travel much faster than mode-II pulses under the same impact amplitude, and the wave speeds of both modes slightly decrease as the impact amplitude increases (Supplementary Fig. 12). With the above observations from Fig. 4c, we demonstrate that the head-on collisions of two mode-II pulses can initiate a nucleation with impact amplitude $A_0 = 0.278$. Then, the new phase grows predominantly along the diagonal at -$45°$. Fig. 4d shows collision of two mode-II pulses propagating along principal axes oriented to one another at $90°$ for $A_0 = 0.24$. Interestingly, we report in Fig. 4e that a mode-I pulse can collide with a mode-II pulse at nearly $135°$ to initiate a nucleation for $A_0 = 0.302$ (note that the pulse of mode I is delayed by

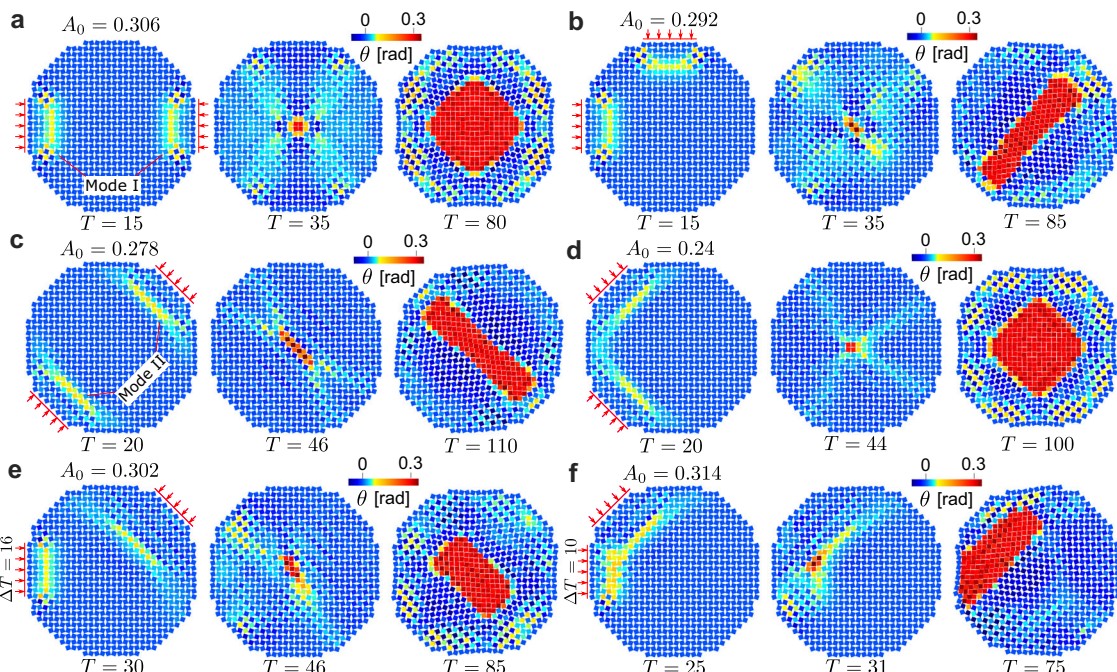

**Fig. 4 | Different collision scenarios. a** Head-on collision of two mode-I pulses with $A_0 = 0.306$. **b** Perpendicular collision of two mode-I pulses, with $A_0 = 0.292$. **c** Head-on collision of two mode-II pulses along the diagonal, with $A_0 = 0.278$. **d** Perpendicular collision of two mode-II pulses, with $A_0 = 0.24$. **e** Collision of a mode-I pulse and a mode-II pulse propagating along directions oriented 135° with respect to one another, with $A_0 = 0.302$. **f** Collision of a mode-I pulse and a mode-II pulse propagating along directions oriented 45° with respect to one another, with $A_0 = 0.314$.

$\Delta T = 16$ to compensate for the speed difference between the two modes). Lastly, Fig. 4f shows the collision of a mode-I pulse and a mode-II pulse propagating along directions oriented 45° with respect to one another for $A_0 = 0.314$ and $\Delta T = 10$. From these various collision scenarios, we note that the spatio-temporal shape of the nucleated phase is affected by where and how the two pulses intersect, which is directly linked to the two impact angles as they dictate the propagation directions and wave characteristics of the two colliding pulses (see Supplementary Note 3 for details). Considering the symmetry of the system, we can control the spatio-temporal shape by changing the impact angles. For example, by moving the left impact shown in Fig. 4b to the right boundary, the new phase will grow predominantly along the other diagonal, at −45° relative to the $x$ and $y$ axes (see Supplementary Fig. 14).

## Discussion

In conclusion, we have experimentally and numerically investigated phase transitions in macroscopic mechanical metamaterials, analogous to classical solid-solid phase transitions in crystals. First, we have experimentally confirmed and numerically corroborated the existence of phase transitions, which can propagate in the form of transition waves in 2D rotating-squares structures. We have identified the fundamental requirements for inducing nucleation, including the energy threshold and the critical nucleus size. More importantly, we have proposed a fundamentally new way via numerical investigations to initiate these phase transitions, i.e., by colliding two soliton-like pulses. This allows nucleation to occur at arbitrary locations in the metamaterial, which may have significant utility in facile control of shape-morphing structures. Although there are a number of practical challenges for experimental observation of nonlinear mechanical waves, such as fabrication errors, material damping, imperfect excitation, etc, the presented approach and the underlying physics could, in principle, be realized experimentally using more advanced manufacturing and measuring techniques. Therefore, this work not only contributes fundamentally to the understanding of nonlinear waves, and particularly how collisions of one type of nonlinear wave can induce formation of another type, but could also open new doors for the design of tunable, shape-transforming, and deployable structures.

## Methods

### Materials and fabrication

In this work, experiments are conducted on building blocks of 2 × 2 elastomeric rotating squares and larger 10 × 10 metamaterials (Fig. 1c). The squares have edge length $d = 12$ mm and are rotated by an angle $\theta_{Lin} = 5°$ with respect to the vertical axis (note that $\theta_{Lin}$ is the initial equilibrium angle without magnets inserted). We print a mold (MakerGear M2, Polylactic acid (PLA)) with cylindrical extrusions of radius $r = 6$ mm at the center (see Supplementary Fig. 1). Adjacent squares are connected via thin hinges of thickness $h = 1.5$ mm. Silicone (Dragonskin 10, Smooth-On, Inc.) is mixed under vacuum using a Speedmixer (FlackTek, Inc), then poured into the mold and cured at room temperature (six hours). After curing, permanent cylindrical magnets (D41-N52 Neodymium Magnets, K&J Magnetics) are embedded at the center to provide attraction between adjacent squares. Note that magnets are not included in the squares along the edges, to prevent unintended phase changes at the boundary squares that can result from boundary effects. Finally, 3D-printed (MakerGear M2, PLA), diamond-shaped trackers are adhered to the surface of each unit to allow tracking of the nodal rotation during dynamic testing.

### Static testing

To characterize the static properties of the sample, we perform quasistatic tensile tests using an Instron model 68SC-5 in displacement control with a displacement rate of 0.02 mm/s. Two aluminum fixtures are used to apply displacement to a specimen comprising four squares (two columns), as shown in Supplementary Fig. 2a. Tensile tests are conducted both with and without magnets.

For tests without magnets (Supplementary Fig. 2b), we embed an aluminum rod at the center of each square. The two ends of the rod maintain alignment via a horizontal slot in the fixture, which allows free

rotation and displacement of each square. Supplementary Fig. 2b, c indicate the locations of the applied force and the direction of rotation of each square. Supplementary Fig. 3 shows the measured force-displacement data (in blue).

As discussed in the main text, we introduce a discrete model to capture the behavior of the prototypes. Based on the discrete model (see schematic in Supplementary Fig. 2d), we can explicitly obtain the force-displacement relationship for a $2 \times 2$ system under tensile loading (the two squares at the bottom are fixed in the vertical direction but free to rotate). The equations of equilibrium for the square highlighted by the red box can be written as

$$
\begin{aligned}
\mathbf{F} + \mathbf{F}_i &= \mathbf{0} \\
2\mathbf{M}_i + \mathbf{r} \times \mathbf{F}_i &= \mathbf{0}
\end{aligned}
\tag{4}
$$

where $\mathbf{r} = \left[ l\sin(\theta_0 + \Delta\theta) - l\cos(\theta_0 + \Delta\theta) \right]^T$, $\mathbf{F}_i = -K_l\Delta u_i\mathbf{e}_y$ and $\mathbf{M}_i = -2K_j\Delta\theta\mathbf{e}_z$ are the longitudinal force and moment of the linkage, respectively. The vertical displacement $u$ (i.e., change in height $H$ defined in Supplementary Fig. 2d) can be expressed as

$$
u = H - H_0 = 2l\cos(\theta_0 + \Delta\theta) - 2l\cos\theta_0 + \Delta u_i
\tag{5}
$$

where $H_0$ is the initial height.

Eq. (4) leads to

$$
\mathbf{F} = F\mathbf{e}_y = -\frac{4K_j\Delta\theta}{l\sin(\theta_0 + \Delta\theta)}\mathbf{e}_y
\tag{6}
$$

For specimens with magnets, we use the Morse potential to empirically capture the magnetic interactions between squares. In this case, the moment from the hinge $\mathbf{M}_i$ becomes

$$
\mathbf{M}_i = -2K_j\Delta\theta\mathbf{e}_z - T_{Morse}(\Delta\theta)
\tag{7}
$$

where $T_{Morse}$ is

$$
\begin{aligned}
T_{Morse}(\Delta\theta) &= \frac{dV_{Morse}}{d(2\Delta\theta)} \\
&= 2\alpha A\left[ e^{4\alpha(\Delta\theta + \theta_0 - \theta_{Morse})} - e^{2\alpha(\Delta\theta + \theta_0 - \theta_{Morse})} \right] \\
&\quad - 2\alpha A\left[ e^{-4\alpha(\Delta\theta + \theta_0 + \theta_{Morse})} - e^{-2\alpha(\Delta\theta + \theta_0 + \theta_{Morse})} \right]
\end{aligned}
\tag{8}
$$

By empirically fitting the experimental data using Eqs. (5)–(7) (red lines in Supplementary Fig. 3), we obtain the following parameters for the discrete model: $K_l \approx 3.958 \times 10^2$ N/m, $K_s \approx 58$ N/m, $K_j \approx 2.5 \times 10^{-4}$ N·m/rad, $A \approx 2 \times 10^{-4}$, $\alpha \approx 8.5$, $\theta_0 \approx 20°$, and $\theta_M \approx 45°$. Then, we can approximate the multistable energy landscape of the hinge and perform the numerical simulation (see Fig. 1c).

## Dynamic testing

To experimentally demonstrate phase transitions, we use a 10-column by 10-row sample on a plastic surface (see Supplementary Fig. 4a). Quasistatic loading is applied to the two vertical hinges connecting the center four squares at the nucleation site. Note, the squares at the edges do not have magnets, to prevent unintended nucleation at the edges induced by boundary effects. Supplementary Fig. 4b, c show a detailed view of the center four squares and friction-reducing feet (MakerGear M2, PLA), respectively. The phase transformation is recorded using a high-speed camera (Photron FASTCAM Mini AX) at 6400 frames per second. Diamond-shaped markers are placed at the center of each square to allow tracking of the rotation and displacement of the squares, using a custom Python script (see Supplementary Fig. 4b). In Supplementary Fig. 5, we plot the experimentally measured angles of the four squares highlighted in inset, showing the transition from the initial phase to the new phase (i.e., Phase R, with $\theta_R \approx 45°$).

## Data availability

Data supporting the findings of this study are included within the paper and its Supplementary Information files. The numerical data underlying the figures and the raw data are generated from numerical simulations using the code available at Ref. 47.

## Code availability

Matlab code for numerical simulations is available at Ref. 47.

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

## Acknowledgements

We gratefully acknowledge support via NSF award number 2041410, AFOSR award numbers FA9550-19-1-0285 and FA9550-23-1-0416, DARPA YFA award number W911NF2010278, and the University of Pennsylvania Materials Research Science and Engineering Center (MRSEC) (NSF DMR-1720530). H.Y. acknowledges the support of KAKENHI (22K14154).

## Author contributions

W.J., H.Y., and J.R. conceived the work. W.J. and H.S. conducted the experiments and analyzed the experimental results. W.J., H.S., and V.T. developed the analytical models. W.J. wrote the code for numerical simulations and analyzed the numerical results. W.J. and J.R. wrote the manuscript. All the authors discussed and reviewed the manuscript. J.R. supervised the research.

## Competing interests

The authors declare no competing interests.
