## [Peer Review File · Nature Communications]

Phase transitions in 2D multistable mechanical metamaterials via collisions of soliton-like pulsesREVIEWER COMMENTS

Reviewer #1 (Remarks to the Author):

Summary:

The manuscript reports observations of phase transitions in 2D multistable mechanical metamaterials that are nucleated in the specimen interior by collisions of soliton-like pulses injected from the specimen boundary. The manuscript discusses the critical energy/size of the nucleating phase, the influence of different collision scenarios defined by the propagation direction of the colliding pulses, and the shape of the resulting low-energy domain. While the reviewer has no concerns with the methods or the validity of the presented results, the reviewer doubts that, in its current form, the manuscript meets the standards of a Nat. Commun. article. However, the reviewer is interested in the authors' response and a revised manuscript.

One important (perhaps, easily fixed) matter is the study motivation. The manuscript introduces and promotes the concept of colliding solitons triggering phase transitions in the specimen's interior that grow to ultimately transform the entire system; however, the reviewer questions the significance of this phenomenon given that phase transitions can be triggered from the specimen boundary (an effect the authors try to avoid) and similarly lead to the entire system being transformed. The authors imply that their method may have utility in the facile control of shape-morphing structures...perhaps the authors can be more specific here and describe a scenario or reference an article(s) where the performance of a shape-morphing structure is improved by a phase transition nucleated from the interior rather than the boundary.

A second reservation the reviewer has is the manuscript's focus. The manuscript discusses (1) the critical energy/size of the nucleating phase, (2) different collision scenarios, and (3) the shape of the low-energy domain acquires during the transformation. In the reviewer's opinion the physics underlying items (2) and (3) are the most novel (especially in the metamaterial context and perhaps beyond that) and should occupy the lion's share of the manuscript's text, rather than item (1) which seems not far enough removed from the authors' earlier work in Refs. [22,24] to warrant the primary focus of a Nat. Commun. article.

In addition, the reviewer has the following comments/questions.

Comments/Questions:

If the whole system transforms from one uniform phase to another, then what is the value of exciting soliton pulses from the boundary? Wouldn't a transition wave stimulated at the boundary achieve the same results?

It will help the reader if the authors clearly state that the "threshold" they are referring to is the critical energy/size of the square region that will nucleate a phase transition. Currently, the authors introduce the search for a threshold in the sentence beginning "Before considering whether collisions..." but it is

not until ~3 paragraphs later in the sentence beginning “The existence of the critical angle” that what threshold is being sought become clearer.

Typo in Ref. 13: “PProc. Nat. Acad. Sci. USA” contains an extra “P”

Reviewer #2 (Remarks to the Author):

This work reports the phase transition wave in 2D multistable mechanical metamaterials in which collisions of soliton-like waves are considered. Experimentally and numerically observation are presented. However, only little extension of previous papers cannot show enough novelties for Nature Communications. The following comments are main reasons that support the reviewer’s attitude.

1.The following papers about solitary waves in mechanical metamaterials with multistable features have already been reported; as well these structures and phenomena are quite similar to this work.

(1) B. Deng, Raney, V. Tournat, K. Bertoldi. Elastic vector solitons in soft architected materials. Physical Review Letters 118 (2017), 204102.

(2) Bolei Deng, Pai Wang, Qi He, Vincent Tournat, Katia Bertoldi. Metamaterials with amplitude gaps for elastic solitons. Nature Communications 9 (2018), 3410.

(3) Bolei Deng, Chengyang Mo, Vincent Tournat, Katia Bertoldi, Jordan R. Raney. Focusing and mode separation of elastic vector solitons in a 2D soft mechanical metamaterial. Physical Review Letters 123 (2019), 024101.

(4) Bolei Deng, Vincent Tournat, Pai Wang, Katia Bertoldi. Anomalous collisions of elastic vector solitons in mechanical metamaterials. Physical Review Letters 122 (2019), 044101.

(5) Bolei Deng, Siqin Yu, Antonio E. Fortea,, Vincent Tournat, Katia Bertoldi. Characterization, stability, and application of domain walls in flexible mechanical metamaterials. Proceedings of the National Academy of Sciences of the United States of America 117 (2020), 31002–31009.

(6) Bolei Deng, Jian Li, Vincent Tournat, Prashant K. Purohit, Katia Bertoldi. Dynamics of mechanical metamaterials: A framework to connect phonons, nonlinear periodic waves and solitons. Journal of the Mechanics and Physics of Solids 147 (2021) 104233.

(7) B. Deng, J. R. Raney, K. Bertoldi, V. Tournat. Nonlinear waves in flexible mechanical metamaterials. Journal of Applied Physics 130 (2021), 040901.

(8) B. Deng, V. Tournat, K. Bertoldi. Effect of predeformation on the propagation of vector solitons in flexible mechanical metamaterials. *Physical Review E* 98 (2018), 053001.

(9) Chengyang Mo, Jaspreet Singh, Jordan R. Raney, Prashant K. Purohit. Cnoidal wave propagation in an elastic metamaterial. *Physical Review E* 100 (2019), 013001.

(10) H. Yasuda, L. M. Korpas, J. R. Raney. Transition waves and formation of domain walls in multistable mechanical metamaterials. *Physical Review Applied* 13 (2020), 054067.

(11) Xinxin Guo, Vitalyi E. Gusev, Vincent Tournat. Frequency-doubling effect in acoustic reflection by a nonlinear, architected rotating-square metasurface. *Physical Review E* 99 (2019), 052209.

(12) Jin L, Khajehtourian R, Mueller J, Rafsanjani A, Tournat V, Bertoldi K, Kochmann D M. Guided transition waves in multistable mechanical metamaterials. *Proceedings of the National Academy of Sciences of the United States of America* 117 (2020), 2319-2325.

(13) Korpas L M, Yin R, Yasuda H, Raney J R. Temperature-responsive multistable metamaterials. *ACS Applied Materials & Interfaces* 13 (2021), 31163-31170.

These suggested papers include both various derivation methods and interesting experiments. Some problems are more difficult and complex than this work.

2. The following studies about the similar structure and solitary wave have been extended to the engineering application, especially for the moving robotic and programmable mechanical metamaterials. But this article is mainly focused on the basic propagation feature.

(1) Ning An, August G. Domel, Jinxiong Zhou, Ahmad Rafsanjani, Katia Bertoldi. Programmable hierarchical Kirigami. *Advanced Functional Materials* 30 (2019) : 1906711.

(2) Xudong Liang, Alfred J. Crosby. Programming impulsive deformation with mechanical metamaterials. *Physical Review Letters* 125 (2020), 108002.

(3) Xudong Liang, Hongbo Fu, Alfred J. Crosby. Phase-transforming metamaterial with magnetic interactions. *Proceedings of the National Academy of Sciences of the United States of America* 119 (2022), e2118161119.

(4) Coulais C, Kettenis C, van Hecke, M Martin. A characteristic length scale causes anomalous size effects and boundary programmability in mechanical metamaterials. *Nature Physics* 14 (2018), 40-44.

(5) Bolei Deng, Mohamed Zanaty, Antonio E. Forte, Katia Bertoldi. Topological solitons make metamaterials crawl. *Physical Review Applied* 17 (2022), 014004.

3. Although the wave equations are delivered here, only numerical calculations are performed instead of the analytical solutions. Previous investigations listed before have obtained different kinds of analytical solutions which are more important than the pure numerical calculations. And the following review paper is suggested to consider in the future:

(1) Dennis M. Kochmann, Katia Bertoldi. Exploiting microstructural instabilities in solids and structures: From metamaterials to structural transitions. *Applied Mechanics Reviews* 69 (2017), 050801.

Reviewer #3 (Remarks to the Author):

This manuscript present interesting results for the propagation of transition waves in two-dimensional multistable metamaterials. Results show that the phase transitions are observed 1) when a cluster of unit cells are quasi-statically rotated and 2) due to collisions of soliton-like pulses. Somewhat limited experimental results show experimental observations of the physical phenomema in the first case. These experiments are followed by more extensive numerical studies using a nonlinear discrete model. The presented research is novel and the simulation results are well presented. The movies of the experiments and simulations are very useful in helping understand the physical phenomena. The study of nonlinear wave propagation in multistable metamaterials is a significant topic that find applications in many different fields.

However, I found that the key weakness of the current manuscript is that there is no rigorous comparison of model and experiments. The model parameters were derived from simple experiments for the force vs displacement curves (Fig. S3), so that good agreement between model experiments and model may be expected in the dynamic case. However, there is a lack of direct comparison. In Fig. 1C, the snapshots are at times in s ; the model simulations in Fig. 1D are in normalized times. Does the model predict similar speed for the transition waves? Previous from the authors (for example Ref. 22) had much more convincing demonstration of the validity of the model for similar systems. The experiments are limited to the quasi-static case and do not include any experiments for the reconfiguration linked to the propagation of two solitons.

Another key limitation of the manuscript is that the model is only a numerical discrete model and this model is used to make empirical observations. There is no effort to develop a theory capable to predicting the cluster size,... It is difficult to understand the generality of the results if only numerical results are presented without a simple underlying theory. Many of the empirical observations are provided without any physical explanation.

Other comments:

1. Abstract: "we experimentally and numerically observe ...": this is technically true, but this sentence overemphasizes the experimental results compared to what is actually presented. 90% of the paper is about numerical simulations, and the abstract should reflect this (unless experiments are more thoroughly analyzed and compared to the model)

2. How does the number of elements influences the response in the case of Fig. 1? Experiments have much few number of unit cells than the model.
3. Fig S7: what is the unit of the horizontal axis (angle in rad or degreee)?
4. What is the "critical nucleus size"? how is it found? Fig. S8 shows not just two different sizes, but two different shapes for the nucleus (rectangular vs cross)? Would a rectangular nucleus (or sufficient large size) be stable in the 2nd case?
5. Why is energy exchange negligible when two tensile impulses of same rotation? What is the physical explanation?

Phase transitions in 2D multistable mechanical metamaterials via collisions of soliton-like pulses

W. Jiao, H. Shu, V. Tournat, H. Yasuda, and J. R. Raney

We greatly appreciate the reviewers' comments and suggestions. A point-by-point response to all the reviewers' comments is given below. The changes made in the revised manuscript and SI are highlighted in blue font.

Response to the first reviewer:

Comment: The manuscript reports observations of phase transitions in 2D multistable mechanical metamaterials that are nucleated in the specimen interior by collisions of soliton-like pulses injected from the specimen boundary. The manuscript discusses the critical energy/size of the nucleating phase, the influence of different collision scenarios defined by the propagation direction of the colliding pulses, and the shape of the resulting low-energy domain. While the reviewer has no concerns with the methods or the validity of the presented results, the reviewer doubts that, in its current form, the manuscript meets the standards of a Nat. Commun. article. However, the reviewer is interested in the authors' response and a revised manuscript.

Response: We thank the reviewer for reviewing our manuscript, and for supporting our opportunity to improve it.

Comment: One important (perhaps, easily fixed) matter is the study motivation. The manuscript introduces and promotes the concept of colliding solitons triggering phase transitions in the specimen's interior that grow to ultimately transform the entire system; however, the reviewer questions the significance of this phenomenon given that phase transitions can be triggered from the specimen boundary (an effect the authors try to avoid) and similarly lead to the entire system being transformed. The authors imply that their method may have utility in the facile control of shape-morphing structures. . . perhaps the authors can be more specific here and describe a scenario or reference an article(s) where the performance of a shape-morphing structure is improved by a phase transition nucleated from the interior rather than the boundary.

Response: We admit that potential applications are long-term and, therefore, speculative to some degree. The emphasis and novelty of this work is the fundamental nonlinear dynamics, and specifically the observation of a new nonlinear behavior (i.e., annihilation of colliding solitons to form a different type of nonlinear wave) which, as far as we know, has not been previously observed in mechanical systems. That said, the presented concepts and results could provide new fundamental insights for the control of the propagation of both vector solitons and transition waves in future mechanical systems. For example, if multiple transition waves are triggered, stationary domain walls can result (H. Yasuda, et al., Phys. Rev. Applied 2020;13:054067), with a distribution of phases depending where and when the transition waves were nucleated. More complex 2D distributions of phases could be achieved using our approach than could be achieved by directly initiating transition waves at the boundary. Second, deployable systems may be heterogeneous, consisting of many structural components. Depending on specific applications, an interior component may need to be multistable and undergo a phase transition. Triggering a phase transition directly from the boundary may not be possible in this case. Finally, initiating phase changes via collisions of vector solitons enables some new capabilities even if the final phase is uniform. For example, it may be desirable for a structure

to undergo the phase transition if and only if a certain threshold of impulses is achieved, i.e., achieving a certain spatiotemporal density of vector solitons in the medium simultaneously. E.g., an aircraft component may change its phase and function once a certain amount of turbulence is encountered. As the reviewer suggested, we have added some text to be more specific about possible applications (see below). However, the primary motivation and contribution of this work is fundamental rather than application-oriented.

The presented method for nucleation of phase transitions could enable new insights for the design of high-dimensional reconfigurable, shape-transforming, and deployable mechanical metamaterials. For example, a deployable structure can be designed to exhibit monostability or multistability at arbitrary locations, and the phase (shape) of the multistable parts, even if they are located in the bulk, can still be controlled (possibly independently) by exciting pulses from the boundary.

Comment: A second reservation the reviewer has is the manuscript’s focus. The manuscript discusses (1) the critical energy/size of the nucleating phase, (2) different collision scenarios, and (3) the shape of the low-energy domain acquires during the transformation. In the reviewer’s opinion the physics underlying items (2) and (3) are the most novel (especially in the metamaterial context and perhaps beyond that) and should occupy the lion’s share of the manuscript’s text, rather than item (1) which seems not far enough removed from the authors’ earlier work in Refs. [22,24] to warrant the primary focus of a Nat. Commun. article.

Response: We thank the reviewer for this comment. We realize that item (1) (i.e., the critical nucleus size and the formation of a critical nucleus via quasistatic loading) was not well-presented in the manuscript. Item (1) is not intended to be a significant focus in and of itself. However, it is necessary to define and discuss the idea of a critical nucleus size and energy threshold in order to establish a baseline energy requirement for the phase transition to occur under any circumstances. Our earlier work that was cited by the Reviewer (i.e., Refs.[22,24]) only considers vector solitons in monostable systems (i.e., single-phase systems), in which phase transitions are not possible. Hence, critical nucleus size was not discussed at all in those previous manuscripts. Nevertheless, we agree that the interesting and novel aspects of the work are items (2-3). In the revised manuscript we have therefore tried to reduce the emphasis on item (1) by moving some content to SI. We have also tried to simultaneously increase the rigor of the discussion of item (1) in the manuscript. Please see new edits in blue font in the revised manuscript and SI.

Comment: In addition, the reviewer has the following comments/questions.

Comments/Questions: If the whole system transforms from one uniform phase to another, then what is the value of exciting soliton pulses from the boundary? Wouldn’t a transition wave stimulated at the boundary achieve the same results?

Response: As mentioned in our earlier responses, we think that this phenomenon is interesting and novel from a fundamental perspective, especially in 2D where there exist a variety of different collision scenarios. Regarding the final result of the transition wave triggering, whether it is triggered directly or remotely via wave collision, we agree that it is the same in the system that we have studied. However, as described in the answer to the reviewer’s second comment, remote triggering by collisions opens up a number of possibilities for control of reconfiguration of more complex structures that consist of multi-stable and monostable regions. In that case, different results from direct excitation of transition waves are expected.

Comment: It will help the reader if the authors clearly state that the ”threshold” they are referring to is the critical energy/size of the square region that will nucleate a phase transition. Currently, the authors introduce the search for a threshold in the sentence beginning “Before considering whether collisions. . .” but it is not until 3 paragraphs later in the sentence beginning “The existence of the critical angle” that what threshold is being sought become clearer.

Response: We agree with the reviewer that this was not clear. In the revised manuscript we now introduce this concept when we first talk about the phase transitions observed in our sample. Moreover, we have performed an energy analysis to better show how a critical nucleus is formed. The new result is documented in the revised manuscript, in which the energy threshold/barrier is better explained.

Comment: Typo in Ref. 13: “PProc. Nat. Acad. Sci. USA” contains an extra “P”

Response: Thank you for pointing this out. It has been fixed in the revised manuscript.

Response to the second reviewer:

Comment: This work reports the phase transition wave in 2D multistable mechanical metamaterials in which collisions of soliton-like waves are considered. Experimentally and numerically observation are presented. However, only little extension of previous papers cannot show enough novelties for Nature Communications. The following comments are main reasons that support the reviewer’s attitude.

Response: We thank the reviewer for reviewing our manuscript. However, our work is not a “little extension of previous papers”. Our study is about inducing phase transitions by collisions of solitons. This is a new fundamental feature of nonlinear waves that has not been observed elsewhere. None of the references cited by the reviewer consider this phenomenon in any capacity.

Comment: 1.The following papers about solitary waves in mechanical metamaterials with multistable features have already been reported; as well these structures and phenomena are quite similar to this work.

Response: We would respectfully ask the reviewer to identify specifically which of our technical claims in the present manuscript “are quite similar to” the references below. This manuscript reports a new nonlinear phenomenon—i.e., the collision and annihilation of solitons leading to new phases in multistable metamaterials. None of the references below discuss or even allude to this possibility. Moreover, the reviewer states that these references describe solitary waves in multistable structures. However, the vast majority of the references (i.e., References 1-9 and 11) do not even describe multistable structures.

Comment: (1) B. Deng, Raney, V. Tournat, K. Bertoldi. Elastic vector solitons in soft architected materials. *Physical Review Letters* 118 (2017), 204102.

(2) Bolei Deng, Pai Wang, Qi He, Vincent Tournat, Katia Bertoldi. Metamaterials with amplitude gaps for elastic solitons. *Nature Communications* 9 (2018), 3410.

(3) Bolei Deng, Chengyang Mo, Vincent Tournat, Katia Bertoldi, Jordan R. Raney. Focusing and mode separation of elastic vector solitons in a 2D soft mechanical metamaterial. *Physical Review Letters* 123 (2019), 024101.

(4) Bolei Deng, Vincent Tournat, Pai Wang, Katia Bertoldi. Anomalous collisions of elastic vector solitons in mechanical metamaterials. *Physical Review Letters* 122 (2019), 044101.

(5) Bolei Deng, Siqin Yu, Antonio E. Fortea,, Vincent Tournat, Katia Bertoldi. Characterization, stability, and application of domain walls in flexible mechanical metamaterials. *Proceedings of the National Academy of Sciences of the United States of America* 117 (2020), 31002–31009.

(6) Bolei Deng, Jian Li, Vincent Tournat, Prashant K. Purohit, Katia Bertoldi. Dynamics of mechanical metamaterials: A framework to connect phonons, nonlinear periodic waves and solitons. *Journal of the Mechanics and Physics of Solids* 147 (2021) 104233.

(7) B. Deng, J. R. Raney, K. Bertoldi, V. Tournat. Nonlinear waves in flexible mechanical metamaterials. *Journal of Applied Physics* 130 (2021), 040901.

- (8) B. Deng, V. Tournat, K. Bertoldi. Effect of predeformation on the propagation of vector solitons in flexible mechanical metamaterials. *Physical Review E* 98 (2018), 053001.
- (9) Chengyang Mo, Jaspreet Singh, Jordan R. Raney, Prashant K. Purohit. Cnoidal wave propagation in an elastic metamaterial. *Physical Review E* 100 (2019), 013001.
- (10) H. Yasuda, L. M. Korpas, J. R. Raney. Transition waves and formation of domain walls in multistable mechanical metamaterials. *Physical Review Applied* 13 (2020), 054067.
- (11) Xinxin Guo, Vitalyi E. Gusev, Vincent Tournat. Frequency-doubling effect in acoustic reflection by a nonlinear, architected rotating-square metasurface. *Physical Review E* 99 (2019), 052209.
- (12) Jin L, Khajehtourian R, Mueller J, Rafsanjani A, Tournat V, Bertoldi K, Kochmann D M. Guided transition waves in multistable mechanical metamaterials. *Proceedings of the National Academy of Sciences of the United States of America* 117 (2020), 2319-2325.
- (13) Korpas L M, Yin R, Yasuda H, Raney J R. Temperature-responsive multistable metamaterials. *ACS Applied Materials & Interfaces* 13 (2021), 31163-31170.

These suggested papers include both various derivation methods and interesting experiments. Some problems are more difficult and complex than this work.

Response: Beyond the superficial similarity of the physical system that we are using as our benchtop experiment, none of these references describe annihilation of colliding solitons, or the formation of transition waves from collisions of a different type of nonlinear wave. I.e., we are using a previously studied system to study new physics. These are fundamental advances that have not been previously examined in the context of mechanical systems by us or (as far as we know) by anyone else.

Comment: 2. The following studies about the similar structure and solitary wave have been extended to the engineering application, especially for the moving robotic and programmable mechanical metamaterials. But this article is mainly focused on the basic propagation feature.

Response: We will take this opportunity to again clarify the purpose and claims of our manuscript. This manuscript is *not* about a “basic propagation feature” of nonlinear waves. We agree that basic propagation of one particular type of wave or another within this structure has already been studied. Instead, our work is specifically about a new type of collision process in nonlinear systems that has not been previously observed.

- Comment:** (1) Ning An, August G. Domel, Jinxiong Zhou, Ahmad Rafsanjani, Katia Bertoldi. Programmable hierarchical Kirigami. *Advanced Functional Materials* 30 (2019) : 1906711.
- (2) Xudong Liang, Alfred J. Crosby. Programming impulsive deformation with mechanical metamaterials. *Physical Review Letters* 125 (2020), 108002.
- (3) Xudong Liang, Hongbo Fu, Alfred J. Crosby. Phase-transforming metamaterial with magnetic interactions. *Proceedings of the National Academy of Sciences of the United States of America* 119 (2022), e2118161119.
- (4) Coulais C, Kettenis C, van Hecke, M Martin. A characteristic length scale causes anomalous size effects and boundary programmability in mechanical metamaterials. *Nature Physics* 14 (2018), 40-44.
- (5) Bolei Deng, Mohamed Zanaty, Antonio E. Forte, Katia Bertoldi. Topological solitons make metamaterials crawl. *Physical Review Applied* 17 (2022), 014004.

Response: The reviewer mentions a few very interesting studies about specific applications of nonlinear waves in similar structures, including programmability and locomotion. However, our study is about a new nonlinear collision phenomenon that has not been reported previously, including in the references provided here by the Reviewer. We do acknowledge that the focus of this manuscript is on fundamental phenomena rather than on applications.

Comment: 3. Although the wave equations are delivered here, only numerical calculations are performed instead of the analytical solutions. Previous investigations listed before have obtained different kinds of

analytical solutions which are more important than the pure numerical calculations. And the following review paper is suggested to consider in the future:

(1) Dennis M. Kochmann, Katia Bertoldi. Exploiting microstructural instabilities in solids and structures: From metamaterials to structural transitions. *Applied Mechanics Reviews* 69 (2017), 050801.

Response: We agree with the reviewer that analytical solutions are preferred wherever possible. In the revised manuscript and supplementary information, we have added additional considerations about analytical solutions in the context of our system. We have devoted efforts to determining analytical solutions whenever possible. For cases where an analytical solution is not available, we have also explained why it isn't. As the reviewer noted, Ref.(3) reports the analytical soliton solutions for similar but monostable systems. We extend the analytical model presented in Ref.(3) to our multistable metamaterial. To derive the analytical solution for solitons, we take the continuum approximation of the discrete model and approximate it with a polynomial the shape of the torsional potential within the energy well around the initial equilibrium. With this analytical model, we can determine whether a soliton solution is available in our multistable metamaterials for a set of parameters. This new analytical investigation has been detailed in new Section 3.2 in the updated SI and mentioned in the revised manuscript accordingly.

We have also thought about the availability of analytical solutions for transition waves in our systems. Unfortunately, we think that analytical solutions for these collisions are not available in the specific system studied here for two reasons:

1)The transition wave triggered either by quasistatic loading or by collisions of solitons propagates from the center of the system, which exhibits anisotropic wave fronts. In this case, it cannot be assumed as 1D propagation of planar transition wave (as was assumed in the previous studies).

2) Planar transition waves are intrinsically not available in our systems, because the vertical displacement in the same phase plane is not uniform due to the shrinking effects associated with the phase change (i.e., the area of the system decreases as it transforms from its initial open phase C to the new closed phase R).

Response to the third reviewer:

Comment: This manuscript present interesting results for the propagation of transition waves in two-dimensional multistable metamaterials. Results show that the phase transitions are observed 1) when a cluster of unit cells are quasi-statically rotated and 2) due to collisions of soliton-like pulses. Somewhat limited experimental results show experimental observations of the physical phenomema in the first case. These experiments are followed by more extensive numerical studies using a nonlinear discrete model. The presented research is novel and the simulation results are well presented. The movies of the experiments and simulations are very useful in helping understand the physical phenomena. The study of nonlinear wave propagation in multistable metamaterials is a significant topic that find applications in many different fields.

Response: We thank the reviewer for reviewing our manuscript, and providing useful comments for us to improve it.

Comment: However, I found that the key weakness of the current manuscript is that there is no rigorous comparison of model and experiments. The model parameters were derived from simple experiments for the force vs displacement curves (Fig. S3), so that good agreement between model experiments and model may be expected in the dynamic case. However, there is a lack of direct comparison. In Fig. 1C, the snapshots are at times in s; the model simulations in Fig. 1D are in normalized times. Does the model predict similar speed for the transition waves? Previous from the authors (for example Ref. 22) had much more convincing demonstration of the validity of the model for similar systems. The experiments are limited

to the quasi-static case and do not include any experiments for the reconfiguration linked to the propagation of two solitons.

Response: We agree with the reviewer that the comparison between simulations and the model were not well-described in the initial manuscript. We have sought to address this in the revision. As the reviewer noted, we had already obtained most of the parameters, except the shear stiffness, for the model from simple tensile tests. For the revision, we conducted additional shear tests to experimentally obtain the shear stiffness (see Fig. S2(e) and Fig. S3(c) in the updated SI). In the simulation, we now use these experimentally-obtained parameters and consider a system of 10×10 squares with geometric properties and boundary conditions identical to the experiments. We superimpose the snapshots from the simulation on top of the experimental images at exactly the same times (with units in s), as shown in new Fig. 1B in the revised manuscript. The comparison shows an excellent agreement between our model and experiments, indicating similar speed for the transition waves (note that the size of the sample is not sufficiently large for calculation of the wave speed). We believe that this provides a more convincing demonstration of the validity of our discrete model.

We acknowledge that, in the current work, the experiments are limited to the quasistatic case, essentially serving the role of validating our numerical model. In separate work, we have experimentally observed collisions of two solitons leading to nucleation of a new phase in 1D systems. However, even in 1D the experiments are very challenging. The structure needs to be long enough that the solitons reach steady-state; yet increasing the length causes prohibitive soliton decay arising from friction and damping. For large 2D structures, this trade-off is even more pronounced, and more sensitive to the choice of parameters. With additional degrees of freedom (i.e., displacement along y axis), additional shear mechanisms, and anisotropy, 2D systems exhibit much richer directional-dependent dynamic properties (see B. Deng, et al., Phys. Rev. Lett. 2019;123:024101) and more complex coupling behavior (this is supported by our numerical simulations; see different collision scenarios). Based on our parametric study presented in the revised manuscript, we find that the parameters need to be carefully chosen to achieve the desired phenomenon. However, many of the key parameters, such as the three stiffnesses of the hinges and those for the Morse potential, cannot be controlled independently, in practice. With the more convincing experimental validation of our model, we hope that the novelty of the presented concept is less affected by the lack of experimental dynamics demonstration. Future efforts could be devoted to experimentally demonstrating the collisions in 2D, with an emphasis on its engineering applications.

Comment: Another key limitation of the manuscript is that the model is only a numerical discrete model and this model is used to make empirical observations. There is no effort to develop a theory capable of predicting the cluster size,... It is difficult to understand the generality of the results if only numerical results are presented without a simple underlying theory. Many of the empirical observations are provided without any physical explanation.

Response: First, we want to highlight that the discrete model is a mathematical model that gives rise to the equations of motion (EOMs), which describe the fundamental dynamic behavior of the physical system. In the revised manuscript, we have performed a theoretical investigation to derive the analytical soliton solutions in the multistable metamaterial. Specifically, we take the continuum approximation of the discrete model and fitting a polynomial to the torsional potential within the energy well around the initial equilibrium. With this analytical model, we can determine whether a soliton solution is available in our multistable metamaterials for a set of parameters. This new analytical investigation has been detailed in new Section 3.2 in the updated SI and mentioned in the revised manuscript accordingly (also see the above answer to a related comment raised by the second reviewer).

To provide a more physical explanation, we have performed an energy analysis on the formation of the critical nucleus induced quasistatically. For the reviewer's convenience, the corresponding text in the revised manuscript is copied below.

To identify the energy threshold for the quasistatically induced phase transition, we calculate the total energy (normalized by $\bar{E} = K_l a^2$) of the 12 squares, including the center four squares (where the loading is applied) and their nearest neighbors, as a function of angle $\theta_0 + \theta_{in}$ (θ_{in} is the input rotation induced by the quasistatic loading). The total energy can be divided into two components. The first is associated with the center four squares (i.e., energy change as they move from Phase C to Phase R). The second is the amount of energy associated with the interface (i.e., the eight nearest neighbors of the center four squares). Interestingly, we observe that the total energy displays an energy barrier E_c at $\theta_0 + \theta_{in} = \theta_c$, as indicated by the blue star in Fig. 1C(i). The existence of an energy barrier implies that, under certain input rotations, the phase transition may be favorable once the energy barrier is overcome during quasistatic loading.

Comment: Other comments: 1. Abstract: “we experimentally and numerically observe ...”: this is technically true, but this sentence overemphasizes the experimental results compared to what is actually presented. 90% of the paper is about numerical simulations, and the abstract should reflect this (unless experiments are more thoroughly analyzed and compared to the model)

Response: As suggested by the reviewer, we have more thoroughly compared the experiments to our model and demonstrated an excellent agreement. However, we agree that the experiments should not be overemphasized in the abstract. We have removed “experimentally and numerically” in that particular sentence. Instead, in the abstract we merely refer to an “experimentally-validated” numerical model.

Comment: 2. How does the number of elements influence the response in the case of Fig. 1? Experiments have much fewer number of unit cells than the model.

Response: We thank the reviewer for raising this important point. To address it, we have performed additional energy analysis to characterize the effects of system size on nucleation when a system is being quasistatically loaded. Based on the energy analysis (see Fig.1C and new edits in the revised manuscript), we find that the effects of system size are very sensitive to the choice of parameters. For the set of parameters that are experimentally derived for the sample, while a nucleus size of 2×2 squares is stable enough in the new phase R to initiate a phase transition for the sample size (10×10), it is not big enough to nucleate the new phase for systems larger than 20×20 (see Fig.1C(i)). In contrast, we have achieved a more robust critical nucleus size of 2×2 for another set of parameters (see Fig.1C(ii)). It is worth noting that physical explanations for the nucleation (or the lack thereof) are provided in the energy analysis.

Comment: 3. Fig S7: what is the unit of the horizontal axis (angle in rad or degree)?

Response: In Fig S7, the units are radians. We have added unit labels for the angles in all relevant figures (note that Fig S7 has been incorporated into Fig. 1C in the revised manuscript).

Comment: 4. What is the “critical nucleus size”? how is it found? Fig. S8 shows not just two different sizes, but two different shapes for the nucleus (rectangular vs cross)? Would a rectangular nucleus (or sufficient large size) be stable in the 2nd case?

Response: We have made this point clearer in the revised manuscript by providing the definition of “the critical nucleus size” when we first discuss the experimental observations of a phase transition in the main text. In the SI section where Fig. S8 is discussed, we have edited the text to make it clearer how to find the critical nucleus size: **By definition, critical nucleus is the smallest cluster size from which the new phase starts to grow. In general, we can start with an initial guess of the critical nucleus size, and perform numerical simulation to determine whether it is the correct one (i.e., whether a phase transition can be observed numerically). If not, we can increase the size and repeat the numerical simulation until the critical size is determined.**

Therefore, as the reviewer suggested, a rectangular nucleus (or sufficient large size) can be stable in the 2nd case, but it may not be identified as the critical nucleus size according to the definition.

Comment: 5. Why is energy exchange negligible when two tensile impulses of same rotation? What is the physical explanation?

Response: By our definition, squares with negative rotation tend to rotate from the initial Phase C to Phase L. For the case considered in the study, this energy barrier cannot be overcome via the collision of two tensile impulses of same (negative) rotation. As indicated in Fig.1A(ii), the energy barrier between Phase C and Phase L is very high and requires very large negative rotation (θ_L). During the collision and before the energy barrier is reached, the rotational kinetic energy gained from the energy exchange is quickly absorbed by the torsional strain energy, especially for the energy landscape between Phase C and Phase L as shown in Fig.1A(ii). Therefore, we observe that the energy exchange does not increase appreciably and is almost negligible when considering two tensile impulses of same rotation.

We have incorporated the above sentences in the updated SI and provided additional physical explanation for another related case (i.e., impulses with opposite rotation) in the revised manuscript.

REVIEWER COMMENTS

Reviewer #1 (Remarks to the Author):

The reviewer thanks the authors for their response to previous comments. In general, the reviewer is satisfied with the state of the manuscript; however, upon this second review, the reviewer has an additional four comments which they believe will improve the manuscript. Comments 1, 2, and 4 are likely to be easily addressed. Comment 3 will likely require more effort. In this case, the reviewer is not expecting an analytical proof, but would appreciate some general guidelines for controlling the phase shape that are derivable from attributes of the metamaterial medium and/or the pulses and/or the pulse collisions.

Comment 1:

The authors cite Refs. 15-18, 23 as examples of research on propagating transition waves in mechanical metamaterials. Many of these works involve one or more of the authors of the current manuscript. To give a better representation of the research being done by several groups in this area, the authors should consider adding the following articles to the list of references on this topic:

Aniket Pal and Metin Sitti, Programmable mechanical devices through magnetically tunable bistable elements, *Proc. Nat. Acad. Sci.* (doi: 10.1073/pnas.2212489120)

Chongan Wang and Michael J. Frazier, Phase patterning in multi-stable metamaterials: transition wave stabilization and mode conversion, *Appl. Phys. Lett.* (doi: 10.1063/5.0152733)

Vinod Ramakrishnan and Michael J. Frazier, Transition waves in multi-stable metamaterials with space-time modulated potentials, *Appl. Phys. Lett.* (doi: 10.1063/5.0023472)

Yujie Zhou et al., Kink-antikink asymmetry and impurity interactions in topological mechanical chains, *Phys. Rev. E* (doi: 10.1103/PhysRevE.95.022202)

Yang Yu et al., Reprogrammable multistable ribbon kirigami with a wide cut, *Appl. Phys. Lett.* (doi: 10.1063/5.0157978)

In addition to Refs. 34-36, in which phase transitions are induced by applying static precompression, the authors may consider adding the following articles:

Audrey A. Watkins et al., Exploiting localized transition waves to tune sound propagation in soft materials, *Phys. Rev. B* (doi: 10.1103/PhysRevB.104.L140101)

Carla Nathaly Villacís Núñez et al., Fractional topological solitons in nonlinear viscoelastic ribbons with tunable speed, *Extreme Mech. Lett.* (doi: 10.1016/j.eml.2023.102027)

Comment 2:

The authors state "Each of these...energy landscape (Fig. 1B)" and "Then, Eq. 1 gives...as shown in the inset of Fig. 1B". In each of these statements, it seems that the Fig. 1B is referenced by mistake and that Fig. 1A(ii) was intended.

Comment 3:

In the abstract, the authors write "Moreover, the rich direction-dependent behavior of the nonlinear pulses enables control of...the spatio-temporal shape of the [nucleated] phase". Given this statement, the reviewer had hoped that, based upon the underlying physics, the manuscript would provide some hypothesis for the nucleation of different shapes. However, all that is provided is a brief note just before the conclusions: "It is also worth noting that these various collisions can lead to nucleation with different shape...". If possible, please provide a physical hypothesis for different impact angles yielding different shapes and (also if possible, even if numerically) provide some testing of this hypothesis in the supplemental material. As it stands, Fig. 4 is a parameter study with lacking insight. With some physical insight or set of principals provided, the "control" of the spatio-temporal shape would seem more deliberate (perhaps, even generalizable)...currently, the "control" seems more haphazard.

Comment 4:

The authors write "...the nucleation can be induced at impact amplitude $A_0 = 0.292$, which is lower than the critical amplitude of a head-on collision...". Can the authors provide an explanation (even a hypothesis) for this result? Perhaps the energy dissipation/dispersion is direction-dependent, affecting pulses at different angles and the amount of energy they can contribute to nucleation at impact.

Reviewer #2 (Remarks to the Author):

Based on the previous comment about this article, it should not be received further consideration.

Reviewer #3 (Remarks to the Author):

The authors have addressed well my comments. The addition of a more quantitative comparison of model and experiments for the quasi-state case is a welcome addition.

in the SI, the text within Fig. S3b should be "with magnets" instead of "without magnets"

Phase transitions in 2D multistable mechanical metamaterials via collisions of soliton-like pulses

W. Jiao, H. Shu, V. Tournat, H. Yasuda, and J. R. Raney

We greatly appreciate the reviewers' comments and suggestions. The changes made in the revised manuscript and SI are highlighted in blue font.

Response to the first reviewer:

Comment: The reviewer thanks the authors for their response to previous comments. In general, the reviewer is satisfied with the state of the manuscript; however, upon this second review, the reviewer has an additional four comments which they believe will improve the manuscript. Comments 1, 2, and 4 are likely to be easily addressed. Comment 3 will likely require more effort. In this case, the reviewer is not expecting an analytical proof, but would appreciate some general guidelines for controlling the phase shape that are derivable from attributes of the metamaterial medium and/or the pulses and/or the pulse collisions.

Response: We thank the reviewer for evaluating the revised manuscript. We also appreciate the reviewer's additional comments which have further improved the manuscript.

Comment 1: The authors cite Refs. 15-18, 23 as examples of research on propagating transition waves in mechanical metamaterials. Many of these works involve one or more of the authors of the current manuscript. To give a better representation of the research being done by several groups in this area, the authors should consider adding the following articles to the list of references on this topic:

Aniket Pal and Metin Sitti, Programmable mechanical devices through magnetically tunable bistable elements, *Proc. Nat. Acad. Sci.* (doi: 10.1073/pnas.2212489120)

Chongan Wang and Michael J. Frazier, Phase patterning in multi-stable metamaterials: transition wave stabilization and mode conversion, *Appl. Phys. Lett.* (doi: 10.1063/5.0152733)

Vinod Ramakrishnan and Michael J. Frazier, Transition waves in multi-stable metamaterials with space-time modulated potentials, *Appl. Phys. Lett.* (doi: 10.1063/5.0023472)

Yujie Zhou et al., Kink-antikink asymmetry and impurity interactions in topological mechanical chains, *Phys. Rev. E* (doi: 10.1103/PhysRevE.95.022202)

Yang Yu et al., Reprogrammable multistable ribbon kirigami with a wide cut, *Appl. Phys. Lett.* (doi: 10.1063/5.0157978)

In addition to Refs. 34-36, in which phase transitions are induced by applying static precompression, the authors may consider adding the following articles:

Audrey A. Watkins et al., Exploiting localized transition waves to tune sound propagation in soft materials, *Phys. Rev. B* (doi: 10.1103/PhysRevB.104.L140101)

Carla Nathaly Villacís Núñez et al., Fractional topological solitons in nonlinear viscoelastic ribbons with tunable speed, *Extreme Mech. Lett.* (doi: 10.1016/j.eml.2023.102027)

Response: We agree with the reviewer that these references are very relevant to this manuscript. We have added them to the revised manuscript.

Comment 2: The authors state "Each of these...energy landscape (Fig. 1B)" and "Then, Eq. 1 gives...as shown in the inset of Fig. 1B". In each of these statements, it seems that the Fig. 1B is referenced by mistake and that Fig. 1A(ii) was intended.

Response: Thank you for pointing this out. We have corrected this typo.

Comment 3: In the abstract, the authors write "Moreover, the rich direction-dependent behavior of the nonlinear pulses enables control of...the spatio-temporal shape of the [nucleated] phase". Given this statement, the reviewer had hoped that, based upon the underlying physics, the manuscript would provide some hypothesis for the nucleation of different shapes. However, all that is provided is a brief note just before the conclusions: "It is also worth noting that these various collisions can lead to nucleation with different shape...". If possible, please provide a physical hypothesis for different impact angles yielding different shapes and (also if possible, even if numerically) provide some testing of this hypothesis in the supplemental material. As it stands, Fig. 4 is a parameter study with lacking insight. With some physical insight or set of principals provided, the "control" of the spatio-temporal shape would seem more deliberate (perhaps, even generalizable)...currently, the "control" seems more haphazard.

Response: We thank the reviewer for this comment. We have performed additional simulations to further demonstrate the potential for controlling the shape of the phases, and have reported these in the revised manuscript. The shape of the nucleation and subsequent propagation (i.e., the spatio-temporal shape of the nucleated phase) is affected by where and how the two pulses intersect. In part, this is because of the anisotropy of the linear and nonlinear (e.g., soliton) wave speeds [see Bolei Deng, et. al. Focusing and mode separation of elastic vector solitons in a 2D soft mechanical metamaterial. Physical Review Letters (123, 024101, 2019) and Section 4.5 in the SI]. This enables the control of the spatio-temporal shape via changing the impact angles. Taking Fig. 4B for example, the intersection of the two mode-I pulses (i.e., along the positive x direction and the negative y direction) results in a nucleation with an initial "\" shape (see the second panel in Fig. 4B). Then, the new phase grows predominantly along the diagonal, at 45° relative to the x and y axes. If we now want to make it grow predominantly along the other diagonal (i.e, at -45° relative to the x and y axes), we can change the two impacts to the right and top boundary, as demonstrated by the additional numerical simulations in the revised SI (Fig. S14 and new edits in the revised manuscript). The corresponding results are presented in Fig. 1.

The effects described above, together with the symmetry of the system, can allow the deliberate control of the spatio-temporal shape, as the reviewer anticipated. We have added this discussion to the revised manuscript (see edits in blue) and performed additional simulations to support it (Section 4.6 in the revised SI) .

Comment 4: The authors write "...the nucleation can be induced at impact amplitude $A_0 = 0.292$, which is lower than the critical amplitude of a head-on collision...". Can the authors provide an expansion (even a hypothesis) for this result? Perhaps the energy dissipation/dispersion is direction-dependent, affecting pulses at different angles and the amount of energy they can contribute to nucleation at impact.

Response: The critical amplitude for nucleation induced via collision depends on several factors, including the propagation distance (e.g., there is dispersion of the pulse in the direction orthogonal to the propagation direction in the 0° and 90° cases), pulse mode, collision angle, energy damping/dissipation (not considered until nucleation occurs), etc. For the specific comparison between the perpendicular collision (Fig.4(A)) and the head-on collision (Fig.4(B)), the only difference is the collision angle (i.e., 90° vs 180°). As observed from numerical simulations, different collision angles can lead to distinct coupling behavior. This can significantly affect the energy exchange between the translational and the rotational kinetic energy, which plays an important role for collision-induced nucleation as discussed in the manuscript.

FIG. 1: Control of the spatio-temporal shape of the nucleated phase. (a) Collision of two mode-I pulses triggered by impacts at the left and top boundary, leading to propagation of the nucleated phase along the diagonal, at 45° relative to the x and y axes. (b) Collision of two mode-I pulses triggered by impacts at the right and top boundary, leading to propagation of the nucleated phase along the other diagonal, at -45° relative to the x and y axes.

Response to the second reviewer:

Comment: Based on the previous comment about this article, it should not be received further consideration.

Response: This reviewer’s objection to our work is founded on a vague claim that the work is not novel. In response to the reviewer’s previous report, we explained why that is incorrect, and asked the reviewer to specifically state which of the technical claims of our manuscript were published previously. We believe this was a simple and reasonable request. The reviewer’s choice to not consider our rebuttal is disappointing.

Regardless, for the sake of thoroughness, we have provided additional details below about the references raised by this reviewer. We have provided a short summary of each, and have clarified the relationship to the current manuscript. In brief, as we described in our previous rebuttal letter, the technical claims of the current manuscript are that (1) we have observed that certain collisions of solitons in 2D mechanical metamaterials lead to annihilation of those solitons; and (2) the annihilation results in a local phase change at the point of collision, which can subsequently affect the rest of the structure via transition waves. While a few references have looked at interactions of solitons (see Ref 4 below), none have done so in a medium capable of changing phase, meaning that annihilation of solitons and subsequent phase changes have not been previously observed (and, in fact, are not possible in those other systems). Similarly, a few references have looked at transition waves in multistable mechanical systems (see Refs 10, 12-13, 16, and 18 below). However, none have considered propagation and collision of solitons in multistable systems, and therefore are not relevant to the claims of this manuscript.

(1) *B. Deng, Raney, V. Tournat, K. Bertoldi. Elastic vector solitons in soft architected materials. Physical Review Letters 118 (2017), 204102:* This work describes 1D propagation of elastic vector solitons in monostable structures. Collisions are not considered. Moreover, the medium is monostable (hence, first-order phase transitions are not possible).

(2) *Bolei Deng, Pai Wang, Qi He, Vincent Tournat, Katia Bertoldi. Metamaterials with amplitude gaps for elastic solitons. Nature Communications 9 (2018), 3410:* This paper describes amplitude gaps that affect the propagation of elastic vector solitons in 1D systems. Again, the structure is monostable, making first-order transitions impossible.

(3) *Bolei Deng, Chengyang Mo, Vincent Tournat, Katia Bertoldi, Jordan R. Raney. Focusing and mode separation of elastic vector solitons in a 2D soft mechanical metamaterial. Physical Review Letters 123 (2019), 024101:* This work is very similar to Ref 1 above, but studies propagation of vector solitons in 2D rather than in 1D. The same caveats apply: Collisions are not considered, and the medium is not multistable.

(4) *Bolei Deng, Vincent Tournat, Pai Wang, Katia Bertoldi. Anomalous collisions of elastic vector solitons in mechanical metamaterials. Physical Review Letters 122 (2019), 044101:* This work is perhaps most relevant to the current manuscript in that it considers collisions of elastic vector solitons. However, these collisions are studied in a 1D structure that is not multistable, hence soliton annihilation and phase transitions are not possible.

(5) *Bolei Deng, Siqin Yu, Antonio E. Fortea, Vincent Tournat, Katia Bertoldi. Characterization, stability, and application of domain walls in flexible mechanical metamaterials. Proceedings of the National Academy of Sciences of the United States of America 117 (2020), 31002–31009:* This paper describes domain walls or disorder that can emerge upon uniaxial compression of mechanical metamaterials based on the rotating-squares mechanism. This work is predominantly about static properties. It does not consider nonlinear dynamics and wave-wave interaction.

- (6) Bolei Deng, Jian Li, Vincent Tournat, Prashant K. Purohit, Katia Bertoldi. *Dynamics of mechanical metamaterials: A framework to connect phonons, nonlinear periodic waves and solitons. Journal of the Mechanics and Physics of Solids* 147 (2021) 104233: This work describes cnoidal wave solutions in 1D systems. Cnoidal waves are a type of nonlinear periodic wave. This work does not describe soliton propagation, collisions, or phase transitions, so it is unclear why the reviewer cited it.
- (7) B. Deng, J. R. Raney, K. Bertoldi, V. Tournat. *Nonlinear waves in flexible mechanical metamaterials. Journal of Applied Physics* 130 (2021), 040901: This is a review paper that mainly summarizes different work conducted by these authors, including Refs. 1-6 and 8. It is irrelevant to the claims of this manuscript.
- (8) B. Deng, V. Tournat, K. Bertoldi. *Effect of predeformation on the propagation of vector solitons in flexible mechanical metamaterials. Physical Review E* 98 (2018), 053001: This is an extension of the authors' previous work on solitons in 1D monostable structures. The focus of this work is on the use of static predeformation (e.g., compression or tension) to affect the propagation of solitons. The medium is not multistable, hence first order phase transitions are not possible.
- (9) Chengyang Mo, Jaspreet Singh, Jordan R. Raney, Prashant K. Purohit. *Cnoidal wave propagation in an elastic metamaterial. Physical Review E* 100 (2019), 013001: As with Ref. 6, this work reports observations of cnoidal waves in monostable 1D structures. Cnoidal waves are irrelevant to this manuscript.
- (10) H. Yasuda, L. M. Korpas, J. R. Raney. *Transition waves and formation of domain walls in multistable mechanical metamaterials. Physical Review Applied* 13 (2020), 054067: This work describes transition waves in 1D multistable structures, and considers how collisions of transition waves can lead to formation of stationary domain walls. Soliton propagation / collision / annihilation is not considered in this work.
- (11) Xinxi Guo, Vitalyi E. Gusev, Vincent Tournat. *Frequency-doubling effect in acoustic reflection by a nonlinear, architected rotating-square metasurface. Physical Review E* 99 (2019), 052209: This paper studies a special nonlinear effect (i.e., frequency-doubling) on the propagation of amplitude-dependent waves in monostable metasurfaces. Collisions are not considered. Moreover, since the system is not multistable, first order transitions are not possible.
- (12) Jin L, Khajehtourian R, Mueller J, Rafsanjani A, Tournat V, Bertoldi K, Kochmann D M. *Guided transition waves in multistable mechanical metamaterials. Proceedings of the National Academy of Sciences of the United States of America* 117 (2020), 2319-2325: Similar to Ref. 10, this paper considers propagation of transition waves in multistable structures, though in 2D instead of 1D. Soliton propagation / collision / annihilation is not considered in this work.
- (13) Korpas L M, Yin R, Yasuda H, Raney J R. *Temperature-responsive multistable metamaterials. ACS Applied Materials & Interfaces* 13 (2021), 31163-31170: This work describes the use of temperature-responsive materials to trigger the propagation of a transition wave in 1D multistable structures, demonstrating temperature-induced propagation of transition waves. Soliton propagation / collision / annihilation is not considered in this work.
- (14) Ning An, August G. Domel, Jinxiong Zhou, Ahmad Rafsanjani, Katia Bertoldi. *Programmable hierarchical Kirigami. Advanced Functional Materials* 30 (2019) : 1906711: This work investigates the 3D deformation and stress-strain response of programmable hierarchical kirigami. Dynamic behavior is not considered in this work, hence it is unclear why the reviewer cited this.
- (15) Xudong Liang, Alfred J. Crosby. *Programming impulsive deformation with mechanical metamaterials. Physical Review Letters* 125 (2020), 108002: This work describes energy conversion processes in a structure capable of storing elastic energy in quasistatic loading and subsequently releasing it via impulsive elastic

recoil. The work does not consider collisions of nonlinear waves.

(16) *Xudong Liang, Hongbo Fu, Alfred J. Crosby. Phase-transforming metamaterial with magnetic interactions. Proceedings of the National Academy of Sciences of the United States of America 119 (2022), e2118161119:* The authors focus on the static and dynamic behavior of the same structure studied in Ref. 15 above, with an emphasis on its potential for energy management and programmable material properties for high-rate applications. The work does not consider collisions of nonlinear waves.

(17) *Coulais C, Kettenis C, van Hecke, M Martin. A characteristic length scale causes anomalous size effects and boundary programmability in mechanical metamaterials. Nature Physics 14 (2018), 40-44:* This paper investigates size effects on static mechanical properties of mechanism-based mechanical metamaterials, such as rotation-based deformation and stiffness. The authors did not cover dynamics, hence it is not relevant to this manuscript.

(18) *Bolei Deng, Mohamed Zanaty, Antonio E. Forte, Katia Bertoldi. Topological solitons make metamaterials crawl. Physical Review Applied 17 (2022), 014004:* The authors leverage the large-amplitude deformation of transition waves to realize crawling locomotion. This work does not consider vector soliton propagation / collision / annihilation.

Response to the third reviewer:

Comment: The authors have addressed well my comments. The addition of a more quantitative comparison of model and experiments for the quasi-state case is a welcome addition.
in the SI, the text within Fig. S3b should be “with magnets” instead of “without magnets”

Response: We thank the reviewer for supporting our revised manuscript, and for pointing out the typo in the text within Fig. S3b. It has been fixed in the updated manuscript.

REVIEWERS' COMMENTS

Reviewer #1 (Remarks to the Author):

The authors have addressed all of my concerns in the revised manuscript and I, now, support its acceptance for publication.

Phase transitions in 2D multistable mechanical metamaterials via collisions of soliton-like pulses

W. Jiao, H. Shu, V. Tournat, H. Yasuda, and J. R. Raney

We greatly appreciate the reviewers' comments and suggestions throughout the review process, which have helped us to significantly improve the content and quality of the manuscript.

Response to the first reviewer:

Comment: The authors have addressed all of my concerns in the revised manuscript and I, now, support its acceptance for publication.

Response: We thank the reviewer again for the helpful evaluation and support of this manuscript.